# Public Green Space Policy Implementation: A Case Study of Krakow, Poland

**Anita Kwartnik-Pruc** *[ID] and **Anna Trembecka**

Faculty of Mining Surveying and Environmental Engineering, AGH University of Science and Technology, 30-059 Cracow, Poland; trembec@agh.edu.pl
* Correspondence: akwart@agh.edu.pl; Tel.: +48-503-151-513

**Abstract:** Green space is essential for the implementation of the idea of sustainable urban development. This paper contains original research on the implementation of local government tasks in the development of public green space. The aim of this research was to analyse the actions taken by the municipal authorities regarding the development of public green space, including the acquisition of real properties, the regulation of their legal status, as well as the adoption of planning and programme documents. The Polish Central Statistical Office data on the public green space of the largest cities in Poland were analysed in order to determine the dynamics of changes. Then, the focus was placed on Krakow, where the authors analysed in detail the distribution and type of urban green space as well as the actions taken by the Municipality to both extend it and to protect it against building development. The criterion of green space accessibility to city residents was indicated as a necessary aspect to be considered in the overall assessment of the existing greenery. The conclusions include the assessment of the actions of the Krakow authorities and the observed trends in the development of public green space.

**Keywords:** green space; sustainable development; land acquisition; regulation of the legal status of green space

## 1. Introduction

The systemic changes that took place in Poland in the 1990s, i.e., after the collapse of communism, led to the re-establishment of local governments along with their basic unit—the municipality. In addition to municipalities, there are now other administrative units in Poland, including counties, covering a dozen or so municipalities, as well as provinces, consisting of several dozen counties. They all form the local government system. Municipalities were authorised to implement a very wide range of public tasks for their own interest and under their own responsibility [1,2]. The performance of public tasks at the level of local government units (at all three administrative levels—the municipality, the county and the province), however, requires that public revenues are divided among them and the central government [3]. Delegating tasks to local governments entails the need to provide funds for their implementation. Therefore, if municipalities are legally obliged to carry out specific tasks (e.g., communal services, transport, primary education, public greenery), the state must support them through subsidies or the transfer of some tax revenues. It is also important to precisely define the tasks and competences of local governments in specific spheres of socio-economic life [4]. One of the tasks of satisfying the collective needs of a community is to provide urban dwellers with a sufficient amount of public green space that has a positive effect on their health and well-being [5–7]. However, municipalities are responsible not only for public greenery [8], but also many other public administration tasks. In post-communist countries such as Poland, the Czech Republic, Latvia, and Romania, the tasks of municipalities are specified in the acts under which these municipalities were established (Table 1). Countries such as the Netherlands or Norway, where the ideas of local government have had the chance to develop for longer,

do not have a specific list of municipal tasks [9,10]. There, the law provides municipalities with independence in determining the needs of a given community. The obligation for municipalities to perform a task that is important from the point of view of the state—such as education or health protection—is imposed by a separate act passed by the parliament.

**Table 1.** Municipal tasks in selected counties. Source: adapted based on [11–14].

| Municipal Tasks | Poland | Czech Republic | Latvia | Romania |
|---|---|---|---|---|
| Municipal roads | + | + | + | + |
| Water supply and sewerage | + | + | + | |
| Maintenance of cleanliness and order | + | + | + | + |
| Landfills and municipal waste disposal | + | + | + | |
| Environmental protection | + | + | + | + |
| Local public transport | + | + | + | |
| Social assistance | + | + | + | + |
| Municipal housing construction | + | + | + | + |
| Public education and culture | + | + | + | + |
| Healthcare | + | + | + | + |
| Marketplace | + | | | + |
| Public greenery | **+** | **+** | **+** | **+** |
| Municipal cemeteries | + | | + | |
| Public order and safety of citizens | + | + | + | + |
| Fire and flood protection | + | + | | + |
| Maintenance of municipal facilities and public utilities | + | + | + | + |

Poland belongs to the group of countries that are still struggling with the post-socialist legacy which, according to [15], includes broad tolerance for inequality, lack of social solidarity and lack of responsibility for the public interest. Part of the existing public green space in post-socialist cities is not protected because it does not have the status of a park, forest or other forms of formal green spaces [16,17]. Failures in securing these areas also result from the reluctance of the municipal authorities to implement the policy of protecting green areas for fear of conflict with developers and loss of revenue [18,19]. It also happens that the Municipality does not have property rights to the land used as public green spaces and must take steps to acquire them or regulate their legal status [20–23]. The limited budget of municipalities, which forces the prioritisation of tasks depending on the needs of a given community, is also a challenge. Despite these difficulties, local authorities in Poland are taking steps to provide their residents with an appropriate amount of green and recreational areas.

*Literature Studies*

Scientists from around the world emphasise the importance of greenery in cities. According to the publication [7], more and more evidence points to the wide-ranging, both long-term and short-term, health benefits of green spaces in cities. Green spaces encourage physical activity [24], and their proximity translates into lower obesity and greater physical activity [25]. According to the research presented in [26,27], urban green spaces promote health and well-being. Public parks contribute to improving the quality of life in the neighbourhood, promote everyday life that is more public, serve as important focal points for neighbourhoods and provide access to nearby nature within a built environment [28]. Societies use city parks in various ways—some for passive recreational activities such as picnicking or relaxing, others for physical activities such as walking or exercising. The publication [29] points to the appreciation of natural values and experienced health benefits as universal attitudes, but also mentions concerns about the general cleanliness and proper maintenance of city parks. It is also pointed out in the subject literature that not only the proximity of green spaces is important, but also the distance necessary to travel to be able to reach them [30]. According to [31,32], the presence of natural areas affects quality

of life in many ways. In addition to environmental and ecological services, urban nature brings important social and psychological benefits by enriching human lives. Allotment gardens also contribute to satisfying the leisure and recreational needs of society [33–35]. The subject of the research additionally covers the issues of the influence of man-shaped nature on the use of space in housing complexes. The proximity of a park increases the value of residential areas [36]. According to [37], the natural shaping of the landscape encourages residents to use outdoor areas more frequently. Based on the conducted research, the authors concluded that spaces with trees attracted larger groups of people and more mixed groups of adolescents and adults than spaces without greenery. These studies suggest that natural elements such as growing trees increase opportunities for social interaction, the monitoring of outdoor areas, and the supervision of children. Therefore, the process of planning new urban green spaces is equally important, as parks play a special role in terms of public space in the planning and development of sustainable land use. Good design helps to position them to perform appropriate cultural and ecological roles [31,38,39]. The publication [40] points to the need to take knowledge about the positive health effects of green space into account in spatial planning. This is exemplified by the research presented in [41], the aim of which was to develop an innovative decision support system for the location of green infrastructure. An issue that has become the subject of research in many countries is the determination of the quantity, distribution and access to green spaces [28,42–46]. The authors of these studies emphasise the need to provide residents with additional space and protect existing green spaces in the light of growing development pressure.

The research studies on green spaces in Poland cover quantitative studies in the country and in selected cities [47–49] as well as spatial analyses related to the accessibility of green spaces [16,17,30,50–52]. Other presented research studies concern ecosystem services [53], allotment gardens [33,34,54], urban planning [55], the management strategy of the Białowieża Forest [56] and the influence of the vicinity of greenery on housing prices [57,58]. The publication [15] discusses the implementation of urban environmental justice in post-socialist cities (also in Poland). Even in the research paper [18], which is a very detailed analysis of 104 publications from around the world regarding green spaces, there is only one that refers to the area of Poland [54]. Therefore, it can be concluded that the activities of municipalities in Poland regarding the implementation of the public green space policy have not been dealt with so far. The presented considerations will fill the knowledge gap and can be used for comparative research in other countries of Central and Eastern Europe.

The research aim of this paper is to answer the question of whether the activities undertaken by local governments at the planning and administrative levels contribute to the development of green spaces in urban areas. This issue is of great significance. According to the literature review, there are no studies assessing the activities of municipalities in the development of public green spaces. This prompted the authors to deal with this research problem. The research may inspire the implementation of additional analyses relating to other public tasks or other cities. According to the authors, the assessment of local government activities formulated in the publication may improve the quality of green space management.

## 2. Materials and Methods

The first research objective was to analyse statistical data on public green spaces in 16 provincial cities in Poland. The data were captured from the local database of the Central Statistical Office. The first research objective defined the:

- Area of public green spaces in individual cities in Poland in 2005–2019;
- Area of public green spaces in relation to the total area of each city;
- Change in the ratio—the area of public green spaces per capita in 2005–2019—for the five largest cities in Poland.

The use of statistical data collected by the state for research purposes is a common procedure due to the ease of access and low cost [18]. However, as far as the data on green

spaces are concerned, there are problems in defining the types of areas occurring in the statistical reports of the Central Statistical Office. The types of green spaces distinguished, such as street greenery or housing estate greenery, do not appear in the cadastral database and cannot be automatically generated and transferred to the Central Statistical Office. Such data, e.g., street greenery, are obtained from area managers. It is frequently the case that the area given in the statistics corresponds to the area managed by a given unit, and not to the actual area (e.g., street greenery) in a given city. Publicly available housing estate greenery may be owned and managed by, for example, housing cooperatives, and then it will not be included in the report prepared by the Urban Greenery Department, which is an organisational unit of the Municipality of Krakow managing green spaces owned by the Municipality. However, since these data are collected in the same way for the entire territory of Poland, they could be used for comparative analysis.

Based on the obtained results, Krakow, the second largest city in Poland, was selected for a detailed analysis due to the significant increase in green spaces between 2005 and 2019, which exceeded 800 ha.

The second research objective was a detailed analysis of the presence of green spaces in Krakow. The analyses were based on data captured from the National Geodetic and Cartographic Documentation Centre. The research specified the:

- Distribution and ownership structure of recreation areas;
- Distribution and ownership structure of forest areas.

As mentioned before, the publication [16] presents the problem of including the selected but not precisely defined greenery in the statistical data. According to the authors [16], the amount of green spaces in the analysed city significantly exceeded the statistical data. However, this study also included agricultural land as green spaces. Agricultural land is largely private property and cannot be equated with public greenery, which is included in the statistical data. The results of the above research pointed to the need of developing the additional categorisation of green spaces in cities that would take access to these spaces into consideration. Therefore, the authors of this research paper analysed the accessible data and prepared a map of the greenery in the city of Krakow that took the accessibility factor into account.

Then, the following were analysed:

- Strategic and planning documents to identify tasks related to green spaces;
- Data on the acquisition of land for public green spaces obtained from the Krakow City Office.

These formed the basis for the analysis of the activities performed by the municipal authorities aimed at increasing the area of commonly accessible green spaces, but also protecting the existing green spaces by acquiring ownership rights to them.

The aim of the analyses was to answer the following research questions:

- What is the distribution and accessibility of green space in the city to its residents? Will this allow areas that require improvement to be identified?
- What actions regarding the development of public green space are undertaken by the Municipality of Krakow?
- Will these actions increase the area of public green space?
- What actions are taken by the Municipality to improve forest cover and what is the effect?
- Are new public green spaces established and developed? Are the existing green spaces protected?

The research method that was used was a case study. The research results were collected, analysed and presented in Section 3.

## 3. Results and Discussion

### 3.1. Analysis of Statistical Data on Public Green Spaces in 16 Provincial Cities in Poland

The first research objective was to analyse the statistical data on public green spaces in Polish provincial cities in the local database shared by the Central Statistical Office. The authors started from analysing their area in each city over the last 15 years, taking into account the area of:

- Parks with walking trails and recreation areas;
- Green squares;
- Street greenery;
- Housing estate greenery;
- Cemeteries;
- Municipal forests.

Figure 1 presents the quantitative area of green spaces in 2005, 2010, 2015 and 2019 in individual cities. It is evident that the dynamics and the direction of changes are different, but in the case of the largest cities such as Warsaw, Krakow, Lodz, Wroclaw and Poznan, the trend is clearly growing. This clearly proves that the authorities of large cities are aware of the need to increase the area of greenery. As the area of public green spaces alone does not give the full picture of the situation, it was concluded that its ratio to the total area of the city should also be examined.

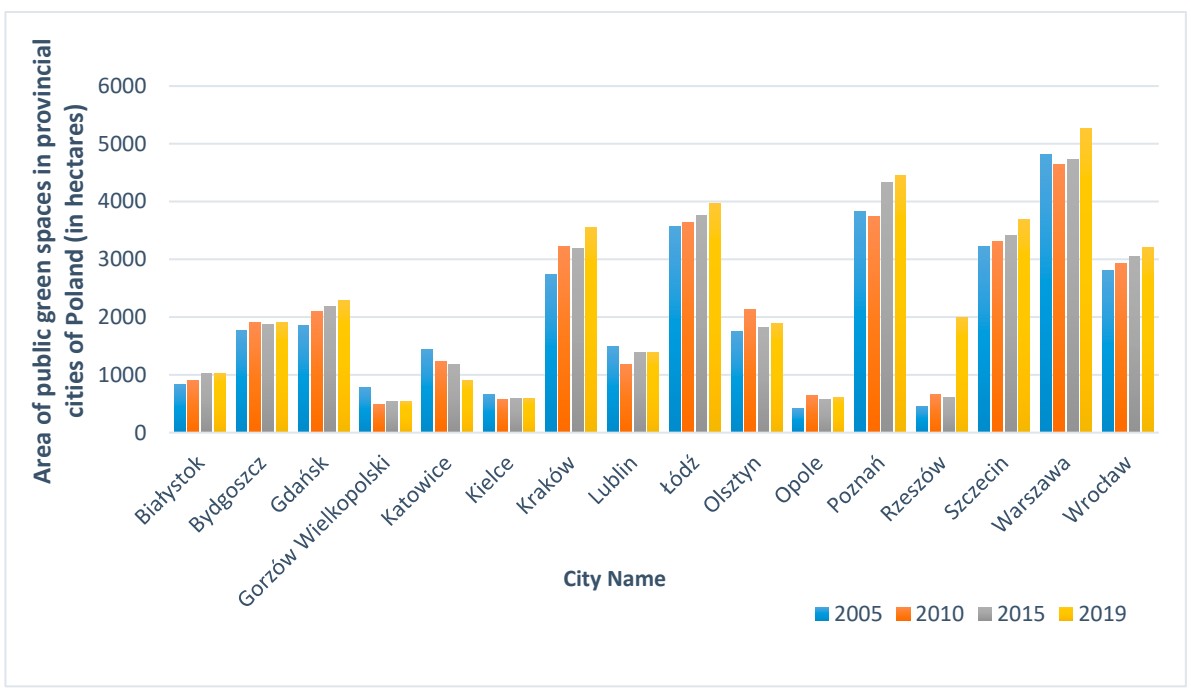

**Figure 1.** Area of public green space (in hectares) in provincial cities in Poland. Source: own study based on the data from the Central Statistical Office.

Figure 2 illustrates the area of provincial cities and the area of public green spaces contained in the database of the Central Statistical Office. The largest Polish city, Warsaw, which is at the same time the capital of the state, covers an area of over 51,000 ha, while public green spaces cover 5255 ha, which is just over 10%. The second largest city, Krakow, which will be analysed in detail later, with an area of 32,685 ha, has 3552 ha of public green spaces, which constitutes 10.9% of its total area. The highest ratio of over 21% is achieved by the relatively small city of Olsztyn, located in one of the country's least populated provinces.

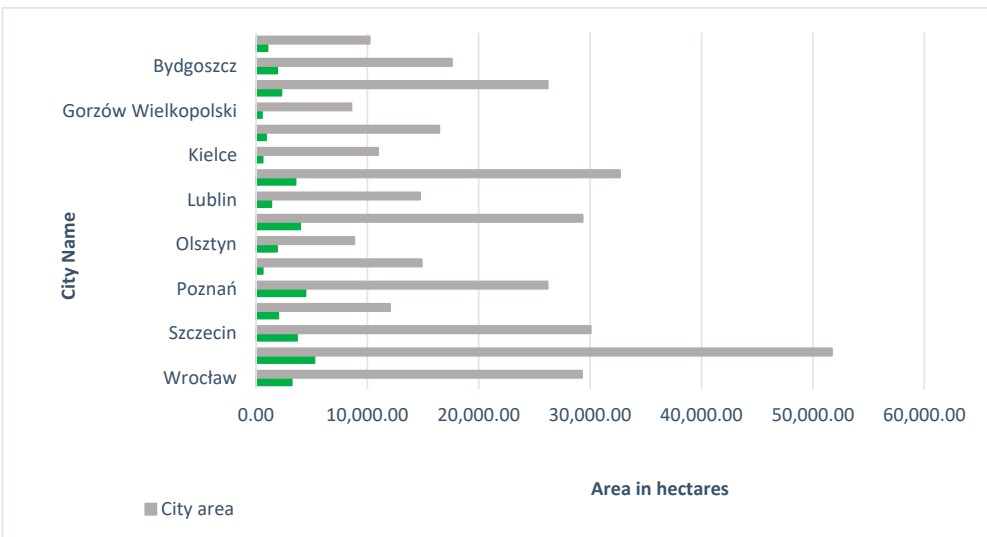

**Figure 2.** Area of public green spaces in provincial cities in Poland as compared to their total area (in hectares). Source: own study based on data from the Central Statistical Office.

Determining the ratio of the area of public green spaces to the total area of the city is important for the assessment of green spaces in the city. However, it was recognised that the number of inhabitants is equally important. Therefore, the next step examined the change in the ratio of the area of public green spaces per capita. Five cities with a population of over 0.5 million were selected for this analysis.

Figure 3 illustrates the changes in the ratio of public green space per capita during the period 2005–2019. It is clearly visible that the worst result was achieved by the largest Polish city and at the same time, the capital of Poland, inhabited by nearly 1.8 million people. In 2005, the ratio was 28 m$^2$/inhabitant, and after 14 years in 2019 it had only increased to 29 m$^2$/inhabitant. The second largest and most populous city in Poland—Krakow—achieved slightly better results. Here, the ratio in 2005 was 36 m$^2$/inhabitant, which by 2019 had increased to 46 m$^2$/inhabitant. Poznan stands out among the largest cities in Poland, as it has visibly improved the quality of life of its inhabitants. Here, the ratio of public green spaces per capita in 2005 was as high as 67 m$^2$, and this ratio had increased to 83 m$^2$ in 2019.

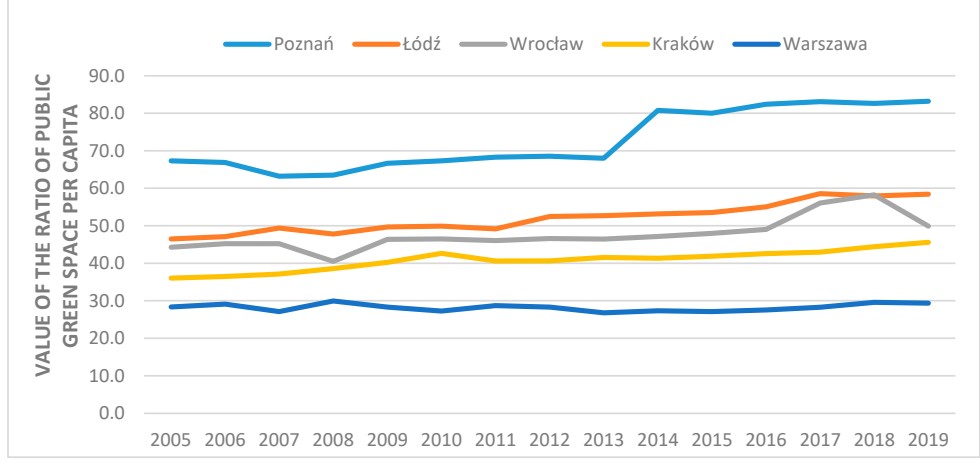

**Figure 3.** Change in the ratio of 1 m$^2$ of public green space per capita during the period 2005–2019. Source: own study based on data from the Central Statistical Office.

Based on the obtained results, Krakow, the second largest city in Poland, both in terms of its area and the number of inhabitants, was selected for a detailed analysis. It is the best known Polish city in the world, which is confirmed by over 12 million visitors in 2017. It is the largest international centre for global business services in Poland [59]. Although it is not the capital of the country (which eliminates the influence of additional financial factors [18]), it is developing dynamically: both the number of inhabitants and the area of public green spaces have increased over the analysed period of 15 years.

### 3.2. Analysis of Krakow in Terms of Spatial Distribution of Public Green Spaces

Krakow, which occupies second position in Poland in terms of its demographic, economic, social and cultural strength, was covered by the detailed analysis. According to the research carried out at the initial stage in Krakow, the ratio of the area of green space per capita is gradually increasing. The city has unique metropolitan functions, and it influences the region, the country, Europe and the world in various ways. It is the "engine" of the Malopolska province. The area of Krakow is 327 km$^2$, and it extends 18 km from north to south and 31 km from east to west.

Issues regarding the municipal greenery are associated with meeting the collective needs of the community and are among the Municipality's own tasks (Article 7 [11]). This standard corresponds to Article 78 of the Nature Conservation Act [60], according to which municipalities are obliged to establish and maintain green spaces and trees in proper condition. The Nature Conservation Act defines green spaces and woodland (Table 2). Pursuant to the regulations discussed above, forests were excluded from trees stands. A forest is a legally different form of greenery, defined in Article 3 of the Forest Act [61]. The presence of forests in Polish cities is something normal. They complement the green fabric of the city.

**Table 2.** Selected definitions of green spaces. Source: adapted based on [60,61].

| Legal Basis | Type of Greenery | Definition |
|---|---|---|
| The Nature Conservation Act | Green spaces | Spaces as developed areas, together with technical infrastructure and functionally related facilities, covered with vegetation, performing public functions. They include, in particular, parks, green squares, promenades, boulevards, botanical gardens, zoological gardens, cemeteries, greenery alongside roads in development areas, squares, historic fortifications, buildings, landfills, airports, railway stations and industrial facilities. |
| | Woodland | Every tree and every shrub if it does not grow in a forest or on a plantation is a tree stand (even a single one, and even more so in clusters), together with the area where it occurs and other components of the vegetation of this area. |
| The Forest Act | Forest | Land with a compact area of at least 0.10 ha, covered with forest vegetation (forest crops)—trees and shrubs and ground cover—or temporarily devoid of it, intended for forest production or constituting a nature reserve or being a part of a national park; forests also include land related to forestry, used for buildings and facilities used for forestry, as well as for forest car parks and tourist facilities. |

The statistical data analysed in Section 3.1 do not provide a complete picture of the green space in Krakow. Therefore, it was decided to perform a detailed analysis illustrating not only its area, but also its distribution. The data were acquired from the National Geodetic and Cartographic Documentation Centre, which includes land registry/cadastral data. The land included in the resource is assigned a type of land use—agricultural land, forests, wooded land, recreation and leisure areas, residential areas. There is no use called "green spaces" that would correspond to the definition contained in the Nature Conservation Act [60]. Figure 4 illustrates the distribution of recreation areas in Krakow, which cover most of the areas indicated in this definition. In accordance with the Regulation on the register of land and buildings [62], recreation areas include undeveloped land and related facilities:

- Recreation centres, playgrounds for children, beaches, parks, squares, green squares;
- Historic areas, including the ruins of castles, strongholds, burial mounds, natural monuments;
- Sports grounds, including: stadiums, sports fields, ski jumps, toboggan runs, shooting ranges, swimming pools, golf courses;
- Areas with entertainment functions, including: adventure parks, amusement parks;
- Zoological and botanical gardens;
- Undeveloped green spaces not classified as forests, wooded or bushy land;
- Family allotment gardens arranged on land that is not suitable for agricultural crops, including forest, wooded and bushy land.

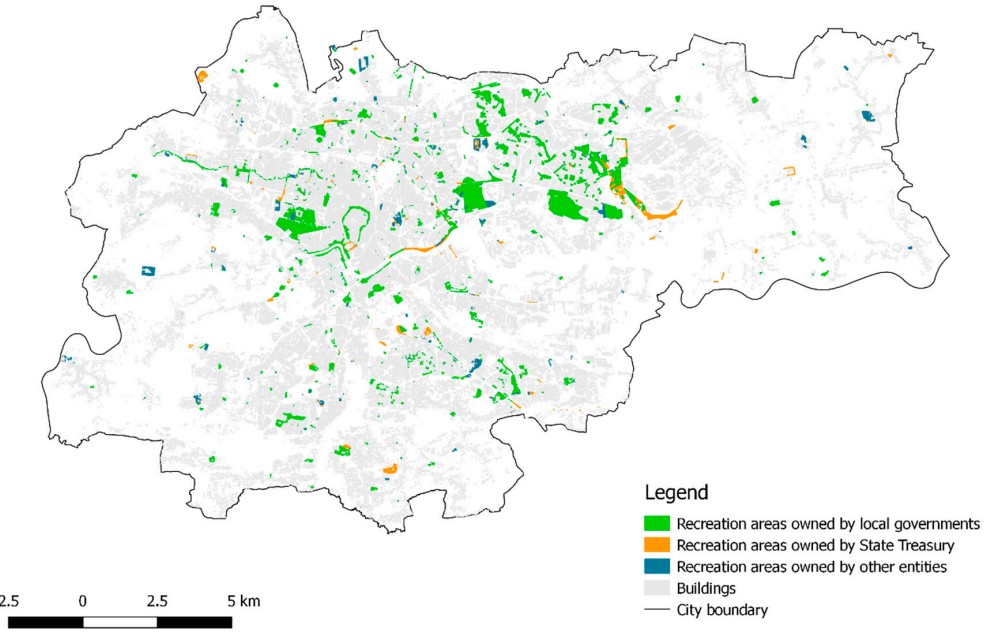

**Figure 4.** Distribution of recreation areas in Krakow. Source: own elaboration.

The areas owned by the local government, State Treasury and other entities are marked in different colours. This land is mostly owned by public entities (as much as 78%).

It is noticeable that these areas are spread throughout the city. The accumulation in the central-eastern part is noteworthy. This is the area of Nowa Huta—a district that was designed and built in the 1960s and 1970s for the employees of the then Vladimir Lenin Steelworks. The central part of the map is the old centre of Krakow with 'Planty'—a park built on the site of the former defensive walls, and on the left there is Błonia Park—the largest area of meadowland (48 ha) in a city centre in Europe. Recreation areas also include numerous allotment gardens. The total area of the gardens exceeds 470 ha of land, which is

about 1.5% of the city's total area [21]. Over the last 10 years, recreation areas have been steadily growing (by 60 hectares during the analysed period of 2010–2020).

Recreation areas are not the only green spaces included in the statistical data of the Central Statistical Office. A significant area is also occupied by trees. Figure 5 illustrates the distribution of forests and afforested land (woodland) in Krakow. Public forests and private forests are marked in different colours.

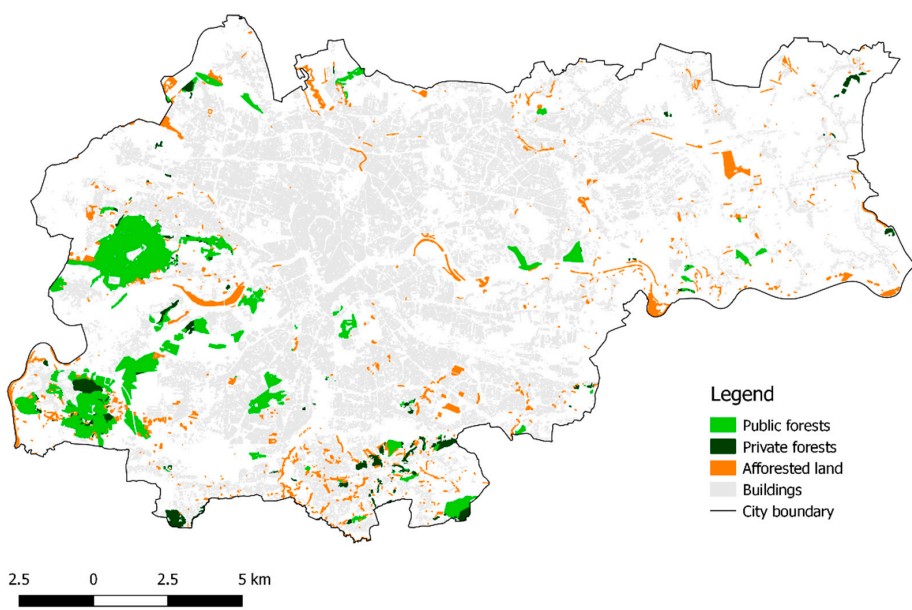

**Figure 5.** Forests and afforested land in Krakow. Source: own elaboration.

The forests illustrated on the map above are the property not only of the Municipality of Krakow but also of the State Treasury, as well as private individuals. However, they are mostly public forests that can be used by all city dwellers. Currently, forests in Krakow cover an area of 1417 ha (as of 1 January 2020), which accounts for 4.3% of the city's total area, and it is one of the lowest ratios in Poland. There is 18 m$^2$ of forest land per capita in Krakow. According to the data that are accessible at the Forest Data Bank and the Central Statistical Office, forest covers 29,6% of Poland's land area. The ownership structure of forests is as follows:

- Municipal forest land—610 ha, i.e., 43% of the total forest area, managed by the Urban Greenery Department and the Municipal Park and Zoological Garden Foundation in Krakow;
- State forests—339 ha, i.e., 24% of the total forest area, fully recognised as protected forests, administered by the Myslenice Forest District;
- Private forests—414 ha, i.e., 29% of the total forest area;
- Forests with other ownership title (undetermined legal status)—54 ha, i.e., 3.8% of the total forest area.

The largest share in the total forest area belongs to municipal forests, followed by privately owned forests and state forests. The statistical data of the Central Statistical Office analysed in Section 3.1 include only municipal forests, and as can be seen from the above data, it appears that almost a quarter of the forests in Krakow are state forests, i.e., public greenery accessible to the general public.

Cemeteries, street greenery and housing estate greenery are also included as public green spaces by the Central Statistical Office. Cemeteries in Krakow, especially the older ones, are in fact an enclave of peace and contain many beautiful trees (location of cemeteries—Figure 6). However, it is difficult to classify them as recreational green spaces

which are defined as the most desirable neighbourhoods in the subject literature [6,28,63]. The same applies to street greenery, the function of which is to protect road users from being dazzled by vehicles coming from the opposite direction, to protect roads against wind and snow, and to protect the adjacent area from excessive noise, air, water and soil pollution. It is also difficult to specifically point to housing estate greenery which, according to the Central Statistical Office, covers over 1000 ha of land. As has already been mentioned above, these data are obtained from area administrators. On the other hand, gated estates, accessible only to residents, are increasingly being constructed in Krakow. Recreation areas located there are not accessible to the public.

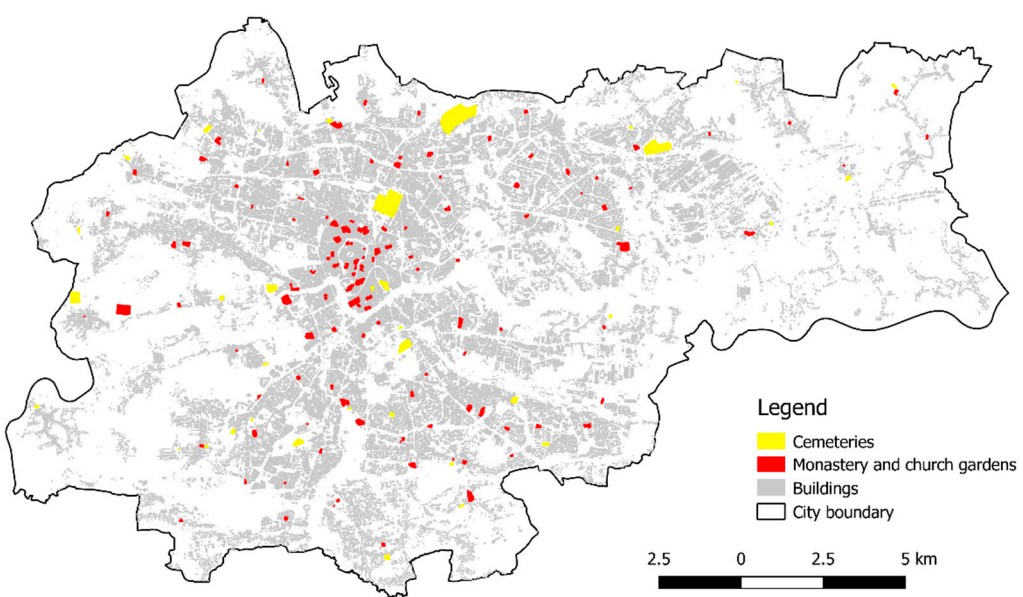

**Figure 6.** Cemeteries, monasteries and church gardens in Krakow. Source: own elaboration.

There is a great diversity of development in the city of Krakow. In the very centre of the old town, there is dense frontage development along the streets, while behind the walls there are often beautiful gardens, especially those that belong to monasteries and churches, marked in red in Figure 6. Unfortunately, these gardens, with two exceptions, are inaccessible to city residents. The case is similar with single-family housing areas. According to Polish law, residential areas include not only land occupied for buildings, but also: yards, access roads, passages, home playgrounds, and leisure areas, wells, tanks, surface utilities, sewage collection and treatment facilities, dustbins, waste landfill sites, small architectural objects, fences, ponds, rock gardens, lawns, flower beds, vegetable gardens [62]. Therefore, there can be a lot of greenery around single-family houses, but the area is considered to be developed.

Analysis of the Accessibility of Green Space

The authors of the publication [16] are right, pointing out that the Central Statistical Office data do not include all green spaces in Polish cities. Similar conclusions can be drawn from the analyses performed above. There are also green spaces in Krakow that are not included in the Central Statistical Office records, e.g., State Treasury forests or greenery within residential areas, but are accessible to residents. Therefore, at the end of the analysis regarding the spatial distribution of green spaces in Krakow, a map was developed (Figure 7), illustrating the existing greenery and the degree of its accessibility. Due to the main research aim, however, the assessment of accessibility, i.e., whether it is possible for the residents to use these areas, was limited. Hence, individual areas were

classified into one of three categories. The areas where only the owners and authorised persons have access were assigned to green spaces inaccessible to the public. These are forest reserves, biological support for watercourses and reservoirs, monastery gardens, shielding and uncontrolled greenery, inaccessible areas next to public utility buildings and greenery of single-family housing developments. Restricted green spaces are areas to which free access is limited due to their nature or legal status. These are private forests, allotment gardens, river parks, fortress greenery, arable crops, areas next to the public utility buildings and sports facilities, as well as the greenery of gated housing estates. Commonly accessible green spaces include those that can be used by all residents (in accordance with the intended use of the area). These are parks, green squares, public meadows, cemeteries, fortress greenery, shielding greenery, uncontrolled greenery, greenery in public spaces, widely accessible greenery next to public utility buildings and sports facilities, widely accessible greenery accompanying multi-family residential buildings, municipal and State Treasury forests.

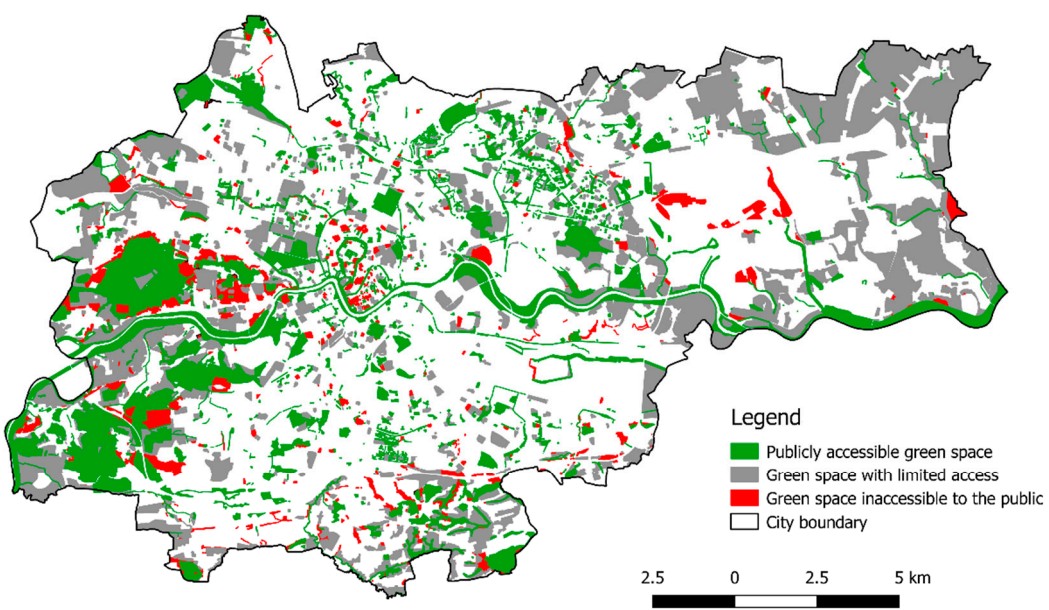

**Figure 7.** Existing green spaces classified according to the degree of accessibility. Source: own elaboration.

Green spaces that are not commonly accessible are marked in red. Their area accounts for 2.9% of the total area of Krakow. The colour grey represents areas with limited access. They account for 18.7% of the city's total area, with arable land accounting for as much as 13.8%. Widely accessible public green spaces are marked in green. These areas in Krakow cover 12.2% of the city's total area.

### 3.3. Analysis of the Actions of the Municipality of Krakow Regarding Development of Public Green Spaces

The public nature of municipal greenery results primarily from the fact that it is to be publicly (i.e., commonly) accessible and is to be owned by a local government unit. The public nature of such objects is determined by the planning document. In Poland, it is a local land-use plan or a decision on establishing the location of a public-purpose investment issued for this type of object. The implementation of tasks regarding the establishment and maintenance of green spaces often requires the municipal local government to purchase real estate owned by private individuals. The acquisition is carried out under civil law contracts or by way of expropriation.

3.3.1. Strategic and Planning Documents that Form the Basis for the Development of Green Spaces in Krakow

The development of green spaces in Krakow is based on strategic documents, which most importantly include the following:

- The Krakow Development Strategy "I want to live here. Krakow 2030" [59];
- Study of Conditions and Directions of Spatial Development of the City of Krakow [64];
- Directions of Development and Management of Green Spaces in Krakow for the years 2017–2030 [65].

**The Krakow Development Strategy** is the city's most important policy paper. Its goal is to make Krakow a city where the slogan promoting the strategy "I want to live here" was both realistic for the inhabitants and a conscious declaration for visitors. The strategy is a key strategic planning document which sets the directions of the city's policy and presents the opportunities and threats for the city in its near and distant future. It was adopted by the Krakow City Council in February 2018, after almost five years of work. Its goal is to make Krakow an open and harmonious metropolis of international importance in the spheres of innovation, science, economy and culture by 2030. This strategy is essential as a reference point for decisions made by the local government administration, strengthening the coherence and predictability of development processes and the standards of strategic city management. As part of the research, the identification and analysis of strategic and operational goals related to the development of green spaces were performed (Table 3).

**Table 3.** Strategic and operational goals related to the development of green spaces. Source: adapted based on [59].

| Strategic Objective IV: Krakow—A City that is Friendly to Live in | |
| --- | --- |
| Operational objective IV.1: Commonly accessible, high-quality public space | Key actions: <br><br> • Revitalisation of existing green spaces and increasing the area of new recreational areas in the city; <br> • Combining scattered green spaces into an integrated system, <br> • Creating and protecting river parks; <br> • Designing green spaces in urban development areas and inside housing estates in the City centre ("pocket parks", courtyards, squares, etc.). |
| Operational objective IV.2: Ecologically sustainable environment | Key actions: <br><br> • Increasing the city's forested area. |
| Goal achievement measure—indicators | • Accessibility of public green spaces for residents (percentage of people living within 300 metres, i.e., within approx. 15 minutes' walk of recreational urban green spaces)—up to 86%; <br> • Area of forests in the total city area (forest area as proportion of the total city area) up to 7.0%; <br> • Number of "pocket parks" (number of small parks up to 0.5 ha) up to 70. |

**Study of Conditions and Directions of Spatial Development of the Municipality** adopted by the resolution of the Krakow City Council CXII/1700/14 of 9 July 2014 [64],

is, after the city development strategy, the most important planning document of the Municipality. The study sought to define the spatial policy of the Municipality, including the local principles of land use planning. The study does not provide grounds for issuing administrative decisions and is not an act of local law; however, its findings are binding for local land use plans, thanks to which the strategic vision presented in this study translates into local law and shapes the development of the Municipality.

After analysing its content, the authors of this paper can also say that this document also supports the process of creating green spaces by assessing the existing natural system and indicating the conditions for its development.

The study of conditions and directions of spatial development of the City of Krakow implements one of the basic local government tasks, i.e., shaping the spatial order, which forms the basis for the development of Krakow. This study formulated the principles and directions of spatial development of the city, as well as detailed guidelines for local plans for shaping the structure of the entire natural system and green spaces. Green spaces in the study were excluded from development and divided into two categories: cultivated greenery and uncontrolled greenery. The basic functions of cultivated greenery include various forms of cultivated greenery (including parks, green squares, river parks), shielding greenery, fortress greenery, greenery of historic premises with building structures, allotment gardens, zoological and botanical gardens. The basic functions of uncontrolled greenery include various forms of uncontrolled greenery, forests, and agricultural land.

The study "**Directions of Development and Management of Green Spaces in Krakow for the years 2017–2030**" is of a conceptual nature and concerns the shaping of the system of public green spaces of the city of Krakow. It takes into account the provisions of local land-use plans that are in force and are available for public review from May 2019 as well as the Study of Conditions and Directions of Spatial Development for the City of Krakow. Moreover, it points to the green spaces not included in the aforementioned documents, but existing or planned to be established. These additional areas, planned to be established, are the directions of development recommended for the Municipality of Krakow in the future, especially in the areas where there are no local land-use plans. The study indicates areas that should be understood as ones in which there is a need to maintain or create green spaces, and the municipal activities related to spatial planning and land resources management should be directed at attempting to meet this demand. A detailed analysis of the document with extensive content was performed, as a result of which the most important objectives of the study were established (Table 4).

In this study, the following existing or planned objects are classified as public green spaces:

- Various types of parks, including 'Planty', river parks, parks located in the area of the Krakow Fortress, spa parks;
- Mounds;
- Town commons;
- Boulevards;
- Squares;
- Green squares and street greenery in road lanes;
- Housing estate green spaces that can be identified and separated from the housing estate area as public parks or squares.

Polish law does not contain any uniform legal regulations regarding the required area of public green space per capita in the city or municipality. The only regulations regarding the area of green space are contained in the Regulation of the Minister of Infrastructure of 12 April 2002 on the technical conditions to be met by buildings and their location [66]. They relate, however, only to the plots covered by construction investments. In the Urban Planning Guide [67] it was recommended that the total area of recreational greenery in cities (excluding the nearest recreation areas in residential areas, which is understood as green spaces on building plots) should not be smaller than 10 $m^2$ of recreational greenery per capita. Based on the recommendations from the Urban Planning Guide and the analysis

of the subject literature, the minimum demand for public green spaces for recreational purposes in Krakow was defined at the level of 10 m$^2$ per capita.

**Table 4.** Basic objectives of the study. Source: adapted based on [65].

| **The main objective** | Defining a coherent, planned and long-term development policy for green spaces in Krakow. |
| --- | --- |
| **Detailed objectives** | • Integration of the scattered green structure into a continuous system of areas connected by walking trails, cycling routes and green spaces;<br>• Maintenance, development and creation of new public green spaces that meet social needs;<br>• Conservation of historic green spaces that are important for the quality of the cultural landscape;<br>• Conservation of valuable natural areas, i.e., spatial and ecological sustainability of the city's development as well as rational management of environmental resources, including water resources;<br>• Raising the standards of the establishment and maintenance of green spaces;<br>• Improving the management of green spaces. |

The directions of the development of public green spaces have been defined in five areas:

- Planning activities;
- Administrative activities;
- Investment activities;
- Conservation activities;
- Educational and promotional activities.

As far as administrative activities are concerned, real estate management as well as the management and administration of green spaces are of great importance. Their main goal is to regulate the legal and formal status of real properties currently managed or maintained by the Urban Greenery Department and real properties newly taken over by this unit for management or maintenance.

3.3.2. Implementation of the Assumed Development Goals for Green Space in Krakow

A very important course of action of the Municipality is the acquisition of land for new investments, which will allow for the continuity of the green space system. The acquisition of land for the benefit of the Municipality of Krakow is regulated by the provision of § 5 Section 1 of the Resolution of the Krakow City Council of 7 May 2003 on the principles of real estate management of the Municipality of Krakow [68]. According to this provision, the acquisition of real estate owned by private individuals is carried out when this property is necessary for the implementation of public purpose investment or the city's own tasks. Therefore, each time it is necessary to prove the legitimacy of acquiring a given plot of land for the Municipality of Krakow's property resource and thus the possibility of using it for the aforementioned purposes, for the implementation of which funds have been reserved in the municipal budget.

The internal procedure for acquiring land for the implementation of municipal tasks stipulates that the necessary condition for taking the aforementioned actions is the existence of a valid land use plan specifying the area intended for acquisition under planned investments, or obtaining a final decision determining the location of a public purpose investment. The Municipality may obtain a decision determining the location of a public purpose investment, e.g., a park located on a property that is not owned by it. According to the current wording of Article 6 of the Real Estate Management Act [69], since 18 November 2015, "dedicating land for publicly accessible: pedestrian routes, squares, parks, promenades or boulevards, as well as their organisation, including construction or reconstruction" has become a public purpose investment. Indications as to the public nature of a given object should also be sought in the provisions of the Act of 16 April 2004 [60] on nature conservation. The catalogue listed in Article 5 clause 21 of this Act covers various forms of green spaces [70]. The public purpose is also to organise these objects, i.e., to establish them (including construction) in a place where they have not existed to date. It may therefore apply to land already in the public domain or land that is not such property yet, but has been dedicated for such a purpose in the local land-use plan, or the location for these purposes has been determined in an appropriate decision on the location of a public purpose investment. Their reconstruction is also a public purpose, which means that it concerns existing buildings.

The acquisition of land is currently carried out under the terms set out in the resolution of the Krakow City Council [68]. The purchase may be implemented at the request of the person who owns the property intended for public green space and is interested in selling it to the Municipality. Acquisition may also be conducted at the request of the organisational unit of the Urban Greenery Department. Procedural steps related to the land purchase are performed by the employees of the City Treasury Department of the Krakow City Office. The acquisition of land takes place in order depending on the planned investments in a given period.

The analysis of the scope of real properties purchased by the Municipality of Krakow since 2003 has demonstrated that a total of 112.2593 ha of land was acquired for investments related to greenery in the period 2003–2019. The acquisition was carried through under civil law contracts after negotiations based on appraisal reports prepared by property appraisers.

The purpose of the acquisition was not only the construction of new parks but also the regulation of the legal status and the extension of existing parks, land afforestation, etc. Apart from sale contracts, real estate exchange and donation agreements were also concluded (in one case, the State Treasury was the donor). Figure 8 and Video 1 (which can be found in the Supplementary Materials) present the area of land acquired for green space within the analysed period, while Table 2 illustrates the purpose of the acquisition.

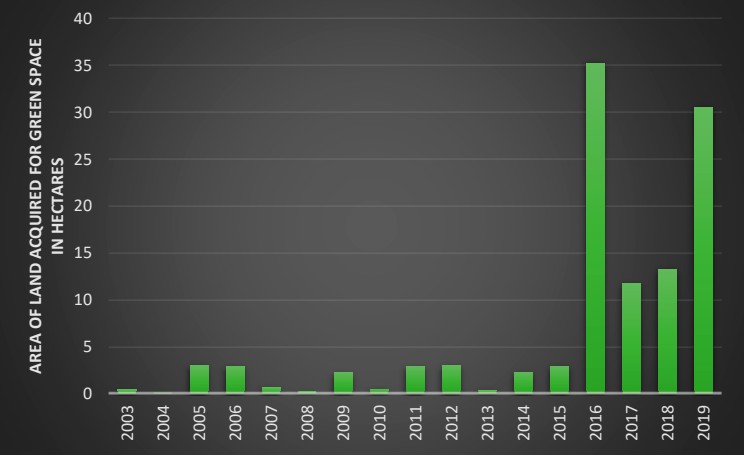

**Figure 8.** Area of land acquired for the benefit of the Municipality of Krakow for green space during the years 2003–2019. Source: own elaboration.

The research shows that since 2016, the area of land acquired by the Municipality of Krakow for green space has significantly increased. Most of the land was acquired in 2016, which resulted from the acquisition of land for Zakrzowek Park. A relatively large area of 30.5809 ha was acquired in 2019, of which approximately 15 ha is occupied by the Borkowski Forest.

As is demonstrated in Table 5, the land with the largest area, i.e., 74.9760 ha, was acquired for the construction of parks of various types (including river parks). The purpose of the acquisition of the land with an area of 1.3746 ha was the regulation of its legal status, which mainly concerned private land used for the existing public greenery. Such actions result in bringing the legal status of land to compliance with the actual state.

**Table 5.** Area and type of land acquired in the years 2003–2019 for the benefit of the Municipality of Krakow. Source: own elaboration.

| Type of Green Space | Area of Land Acquired for the Benefit of the Municipality of Krakow (ha) |
| --- | --- |
| River parks | 23.1267 |
| Other parks | 51.8493 |
| Regulation of legal status existing parks | 1.3746 |
| Public green areas | 18.6056 |
| Forests | 15.0308 |
| **Total** | 112.2593 |

According to the data as of December 2019, there were 47 public city parks in Krakow covering an area of 446.9994 ha (Figure 9) and Spa Park in Swoszowice, which is publicly accessible but managed by a health resort (the area that is currently accessible amounts to 4.5130 ha). Altogether, it is an area of approximately 451.5 ha throughout the city. The area of public green spaces with a recreational function, such as parks and squares, was 618.48 ha at the end of 2019. According to the Urban Greenery Department, there are currently as many as 49 parks in Krakow—the most recent two are Stacja Wisła Park and Reduta Park with an area of 7.8 ha, located among completely new high-rise buildings. The purchase of private land by the Municipality of Krakow saved them from being fully developed with residential buildings.

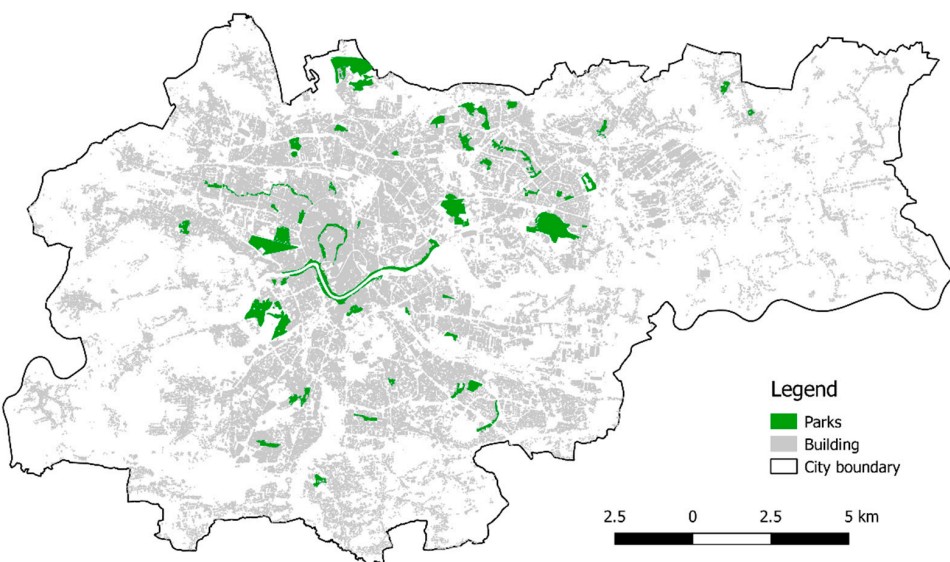

**Figure 9.** Parks in Krakow. Source: own elaboration.

In 2018, in Krakow, it was decided to establish 18 pocket parks—one in each district. Such parks are important for local communities [71] and their function can be adapted to the needs of local residents [26]. The inhabitants of each district decided their location. It is evident that the city authorities are aware of the need to cooperate with residents.

Another noteworthy initiative carried out periodically since the spring of 2018 is called Krakow Residents Parks and involves city residents planting trees in the spring and autumn. The Urban Greenery Department provides space and saplings. In this way, several hundred trees have, to date, been planted in Krakow [72].

As part of the implementation of objective IV.2, "an ecologically sustainable environment", the "county programme for increasing forest cover in the city of Krakow for the years 2018–2040" was adopted [73]. It is part of one of the main objectives of the national forest policy. The basis for the implementation of this objective was the "national programme for increasing forest cover", the aim of which was to ensure conditions for increasing the forest cover to 30% of the total area of the country by 2020, the optimal distribution of afforestation, setting ecological and economic priorities as well as implementation instruments [74].

The programme sets out the rules and conditions for increasing the forest area in the Municipality of Krakow, ultimately at a level of no less than 8% of the Municipality's total area. This is in line with the research presented in [75,76]. The aim of the programme, the course of which is defined in four stages, was to achieve the declared increase in forest area in the Municipality of Krakow by

- Dedicating 574 ha of non-forest land to be changed into the "forest" land-use category;
- Dedicating 856 ha of non-forest land for afforestation and changing to "forest" land-use category in the longer term, so that by 2040, forest cover is not lower than 8% of the Municipality's area.

The monitoring of the programme implementation will be based on the annual reporting of the area afforested in a given year (ha), on the area changed to forest land use category in a given year (ha), and at the end of each of the four stages, on the calculation of the forest cover index (%). Forest cover is expected to increase from a base value, i.e., 4.3% of the area of the Municipality of Krakow, to the target value of 8%.

The effect of the measures that have already been taken to increase forest cover is afforestation in the city:

- In 2018, nearly 13,000 saplings were planted on approximately 3 ha of land;
- In 2019, 120,000 saplings were planted on approximately 23 ha of land;
- In 2020, approximately 74,000 saplings were planted in spring on approximately 15 ha of land.

The research shows that activities undertaken by local governments at the planning and administrative levels contribute to the development of urban green spaces. This is evidenced by the implementation of the key actions included in the Krakow Development Strategy and "Directions of Development and Management of Green Spaces in Krakow for 2017–2030", aimed at the following:

- Increasing the area of new recreational spaces in the city;
- Merging scattered green spaces into an integrated system (which requires acquisition of new land in complexes);
- Establishing and protecting river parks;
- Designing green spaces in urban areas ("pocket parks");
- Increasing the area of forest in the city.

The research confirms that these actions undertaken by the Municipality of Krakow resulted in a gradual increase in the area of land acquired for green spaces, from 0.4342 ha in 2003 to 30.5809 ha in 2019. The largest increase in the area of the acquired land has been observed since 2016. In this year, for example, 20 hectares of a park existing on private land were purchased.

The adoption of programmes obliges municipalities to allocate part of the funds in the budget to green spaces. It also frequently defines areas that require action. Therefore, municipalities are mobilised to effectively implement the planned objectives, e.g., to achieve the assumed assessment indicators. The disadvantages include, e.g., the necessity to adopt changes to the programmes by the Municipal Council, should the need arise to move the location of a public green space.

The literature indicates that the use of green spaces is influenced by various factors: financial resources, economies and markets, political leadership and the availability of land [18]. The presented research focuses on the analysis of strategic documents of the Municipality of Krakow and the analysis of processes of land acquisition to establish, improve and protecting green spaces. The actions assessed directly affect the creation of an optimal system of urban green space. It will definitely be worth repeating the research at some time in the future in order to assess the dynamics of the changes. It would also be interesting to expand the research area to, e.g., bicycle paths or to compare the activities of local governments in other cities.

## 4. Conclusions

When analysing the state of public green space in the city, one should consider not only quantitative data, but also the distribution and type of individual areas in the city structure and their accessibility to residents.

In the analysed area of the city of Krakow, the successive increase in green space was possible thanks to the actions of the local government authorities who:

- Created strategic and planning documents defining a coherent, planned and long-term policy for the development of green spaces in Krakow;
- Successfully implemented the process of land acquisition for greenery, which required a number of actions as well as the implementation of administrative and planning procedures;
- Provided financial resources in the budget.

The research presented above identified the challenges faced by local government in the management of green areas. These are:

- The even distribution of green spaces in cities;
- Increasing the accessibility of green spaces with limited or excluded access for residents;
- The gradual expansion of public green spaces, especially in intensively built-up areas;
- Maintaining existing green spaces in good technical condition;
- Adapting the nature of green spaces to the needs of local residents.

According to the authors, other factors that were beyond the scope of this study but determine the development of green space should also be examined. These include, for example, funds that may be allocated by municipalities for this purpose. Conclusions from such studies, in combination with green space development plans, could determine the scale of the need for financial resources and encourage looking for other sources of income, e.g., through subsidies from the State Treasury.

The analyses performed based on the city of Krakow could become:

- A reference point for the assessment of the state of greenery in other Polish and foreign cities of comparable size;
- An inspiration to improve the actions of local governments regarding the development of public green space;
- Useful for the assessment of the implementation of other public tasks.

The study contributes to broadening global knowledge on the implementation of public tasks regarding greenery development performed by municipalities in Poland.

**Supplementary Materials:** The following are available online at https://www.mdpi.com/2071-105 0/13/2/538/s1, Video S1: Land acquired for green space in Krakow during the years 2003–2019.

**Author Contributions:** Conceptualization, A.K.-P. and A.T.; methodology, A.K.-P. and A.T.; formal analysis, A.K.-P.; investigation, A.K.-P. and A.T.; resources, A.K.-P. and A.T.; data curation, A.K.-P. and A.T.; writing—original draft preparation, A.K.-P. and A.T.; writing—review and editing, A.K.-P. and A.T.; visualization, A.K.-P. All authors have read and agreed to the published version of the manuscript.

**Funding:** The article was prepared under the research subvention of AGH University of Science and Technology No. 16.16.150.545.

**Institutional Review Board Statement:** Not applicable.

**Informed Consent Statement:** Not applicable.

**Data Availability Statement:** This study analyzed publicly available datasets. This data can be found here: [https://stat.gov.pl, https://msip.krakow.pl, https://zzm.krakow.pl].

**Conflicts of Interest:** The authors declare no conflict of interest.

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
