# Peer review of "Public Green Space Policy Implementation: A Case Study of Krakow, Poland"

_sustainability, doi:10.3390/su13020538_

Round 1
Reviewer 1 Report
The article is well organized, easy to be read and understand. The theme of the article is very ”popular” and the article title creating an initial good impact.
For the future, the research can be extended with supplementary geospatial analysis.
Reviewer 2 Report
#Comments to authors
It is a great opportunity to review a manuscript entitled "Implementation of Local Government Tasks Regarding the Development of Public Green Spaces. City of Krakow – Case Study Analysis." The authors assessed the local government tasks to provide public green spaces for people's wellbeing and ecosystem protection.
This is an interesting academic paper; however, the authors should address the following concerns:
- The introduction:
- Figure 1 is very colourful and messy. We suggest the authors redesign it in a professional format.
- The introduction is very broad and generic. In the first paragraph, the authors talked about the local government's tasks, but the authors did not find out what problems local governments were facing. In the second paragraph, the authors tried to describe the advantages of green space that have already been known. It would be interesting if authors focused on the challenges that the local governments in Poland were facing to provide/manage green space as the topic was on the implementation of the local government tasks to provide public green space.
- The objectives were not clearly defined.
- The method and materials
- Since the authors used secondary data collected by the Central Statistical Office, it is important to describe how data were collected. In addition to the data collection methods, it is also essential to explain the data's accuracy and validity.
- Results
- The way authors wrote in the results, and the conclusion is very confusing. In the result section, the authors should write only results +discussion (if the authors wish to combine results and discussion together). Whereas, in the conclusion section, it is academically important to conclude what the study found/produced+ policy recommendations and future research direction (if any).
- Conclusion
- May the authors describe critical challenges and opportunities of the local governments in providing/managing public green spaces in Poland?
- How can the findings of this study contribute to the resolution of the challenges, and strengthen the performance of the local governments in terms of public green space management?

Reviewer 3 Report
See attached PDF

Reviewer 4 Report
This is an interesting case study on the changes in public green spaces in Krakow, Poland. However, the contents are mainly a retrieval of factual data and information from government documents, rather than an analysis.
The thesis of the study is stated to be the statement that "the development of green space in urban areas is the result of actions undertaken by local governments at the planning and administrative levels", and "The authors analysed in detail the distribution and type of urban green space as well as the actions taken by the municipal authorities to both extend it ..." (lines 118-128)
However, there is no actual analyses or evidence on the association between government actions and the changes in public green spaces.
I suggest focusing on comparing at least two cities to contrast with the differences in government actions and their consequences in the differences in the development of public green spaces.
Furthermore, a video (or a continuous series of maps selectable by a time bar from 2003-2019) showing the temporal-spatial changes in government actions (such as buying private land) and temporal-spatial changes in public green spaces can help visualise the relationship between government's actions and the increases in public spaces.
More importantly, how can the study be relevant to the international audience of the journal shall be addressed. What are the lessons learned from this case study? For example, if two or more cities' cases study can be contrasted with, then the results may be able to illustrate the importance of governments' actions on the development of public green spaces. Or if the reasons why the rapidly increasing area of land acquired (Table 1) can be analysed and compared with other cities, then it may be a lesson learnt.
The strategic and planning documents (Krakow Development Strategy, Study of Conditions and Directions of Spatial Development of the Municipality, and Directions of Development and Management of Green Spaces in Krakow for the year 2017-2030) are described but not analysed. If the argument is that it is the written strategies that makes a difference in the development of public green spaces, then it shall provides strong evidence on the changes of the government's actions before and after the publication of the strategies. It is because something written on a document may not be 100% implemented.
Lastly, since the study method is a case study of Krakow city, the general information of the areas of public green spaces in each city of Poland (Diagram 1), the areas of city areas (Diagram 2), and the per capita public green space of 5 cities (Diagram 3) can be simplified and combined, and put in the introduction section. Diagram 4 (population) is not necessary. They should not be included in the Results section.
Furthermore, all the diagrams are unclear and without legends or notes. For example, the source of Diagrams 1 & 2 seems to come from the CSO, but it is stated as "own study" or "own elaboration".
Author Response
Please see the attachment. I have prepared a video illustrating the process of the purchase of land by the Municipality of Krakow for green spaces in 2004-2019, but I can’t send it with a response – only Word/PDF file.

Round 2
Reviewer 2 Report
# Comments to authors
I could see the author’s efforts in revising the original manuscript. The quality of the revised version is much better. I satisfy with the current version. However, there are some minor issues that the authors should consider:
- Citation of your owned work. Page 11, table 2, page 18, table 3, and page 19, table 4. For example, you cited your owned work, Ref.64, twice in Table 4. Please revise it.
- Format and structure of the manuscript required by the journal. You need to check the guidelines carefully in order to ensure that the current format and structure are consistent with the Sustainability’s requirements.
- The conclusion should be the end of your manuscript. Avoid any details describing the results again or attaching figure/table. The conclusion should consist of 3 important aspects:
- Conclude your results in summary
- Policy implication/recommendation
- Future research direction that will complement your work or what has not been done and is important to conduct the research in the future
Reviewer 3 Report
Comments attached (SUSTAINABILITY_976659-R2 Review comments for authors.pdf)

Round 3
Reviewer 3 Report
This manuscript has been significantly improved since its initial submission, with most suggestions now incorporated in the current version. There are still a few typos (see Table 5 and References) but these are now mostly copy editing issues in my view.
The presentation, particularly the written expression, grammar, and at times, word choice, are improved and considered acceptable for an academic journal publication. The research purpose, questions, findings, and interpretation are now clear to the reader. The authors are commended for their efforts detailing the procedures for acquiring land. This is a fundamental step in greenspace provision that is not widely appreciated or understood. There is now a more balanced approach to quantity of Figures (Diagrams and maps) now supported with Tables. The Conclusions section, while short and reliant on bullet points, describes some of the research limitations, emerging questions for further research to examine, and suggests the implications of these findings at local, national, and global scales.
The research undertaken is interesting, should make a worthy contribution to scholarly knowledge on the demonstrated strengths and the type application of greenspace planning instruments (as governance tools) are at play in a local context, including the challenges faced by post-communist countries emerging from such political regimes.
Author Response
Dear Sir or Madam,
We would like to thank you for accepting our manuscript.
Table 5 and the Conclusions have been corrected.
Yours faithfully,
Authors